# Taxonomic Diversity and Functional Traits of Soil Bacterial Communities under Radioactive Contamination: A Review

**DOI:** 10.3390/microorganisms12040733

**Published:** 2024-04-03

**Authors:** Elena Belykh, Tatiana Maystrenko, Ilya Velegzhaninov, Marina Tavleeva, Elena Rasova, Anna Rybak

**Affiliations:** 1Institute of Biology of Komi Scientific Centre, Ural Branch of Russian Academy of Sciences, 28 Kommunisticheskaya St., Syktyvkar 167982, Russiavellio@yandex.ru (I.V.); elrasova@mail.ru (E.R.); 2Department of Biology, Institute of Natural Sciences, Pitirim Sorokin Syktyvkar State University, 55 Oktyabrsky Prospekt, Syktyvkar 167001, Russia

**Keywords:** radioactive contamination, ionizing radiation, bacterial communities, taxonomic diversity, resistance, adaptation, functionality

## Abstract

Studies investigating the taxonomic diversity and structure of soil bacteria in areas with enhanced radioactive backgrounds have been ongoing for three decades. An analysis of data published from 1996 to 2024 reveals changes in the taxonomic structure of radioactively contaminated soils compared to the reference, showing that these changes are not exclusively dependent on contamination rates or pollutant compositions. High levels of radioactive exposure from external irradiation and a high radionuclide content lead to a decrease in the alpha diversity of soil bacterial communities, both in laboratory settings and environmental conditions. The effects of low or moderate exposure are not consistently pronounced or unidirectional. Functional differences among taxonomic groups that dominate in contaminated soil indicate a variety of adaptation strategies. Bacteria identified as multiple-stress tolerant; exhibiting tolerance to metals and antibiotics; producing antioxidant enzymes, low-molecular antioxidants, and radioprotectors; participating in redox reactions; and possessing thermophilic characteristics play a significant role. Changes in the taxonomic and functional structure, resulting from increased soil radionuclide content, are influenced by the combined effects of ionizing radiation, the chemical toxicity of radionuclides and co-contaminants, as well as the physical and chemical properties of the soil and the initial bacterial community composition. Currently, the quantification of the differential contributions of these factors based on the existing published studies presents a challenge.

## 1. Introduction

The soil is rich in microorganism diversity [1,2], as evidenced by the presence of 10^6^–10^10^ prokaryotic cells per gram, including bacteria and archaea [3,4]. The activity of these microorganisms, often referred to as soil engineers, plays a crucial role in maintaining the stability and health of the soil environment by significantly contributing to soil biochemical processes [5,6]. Additionally, these microorganisms play a vital role in maintaining the material and energy cycles in the biosphere [7].

Soil contamination has adverse effects on all organisms inhabiting the soil, particularly bacterial communities. Prokaryotic organisms, characterized by features such as a high rate of growth and reproduction, and horizontal gene exchange, exhibit a superior and faster adaptation to changing environments compared to other organisms. A specific group of bacteria known as extremophiles are exceptionally well-adapted to survive in extreme conditions, including environments with elevated [8] concentrations of heavy metals (HMs), extreme temperatures, and high levels of radiation exposure [9,10,11].

Soils in various regions exhibit elevated concentrations of radionuclides (RNs) due to both the natural conditions and anthropogenic activities, posing a potential hazard to living organisms. The causes of radioactive contamination can be attributed to several factors: areas with increased natural radioactivity [12,13,14,15,16], technologically enhanced natural radioactivity [17,18,19,20,21,22,23,24,25], radiation accidents [26,27,28,29], nuclear tests [30,31,32,33], and oil pollution [34,35,36,37,38].

In turn, the activity of microorganisms plays a significant role in influencing the chemical form, mobility, and toxicity of radionuclides in soils, thereby either impeding or promoting their incorporation into biological migration pathways [39,40,41,42,43,44,45]. This is why bacteria, together with microscopic fungi, are used as a key component in complex biostimulants for remediating contaminated soils, particularly in cases of polymetallic and radionuclide pollution [46,47,48].

In a global context, the radioactive contamination of the soil poses a long-term radiation hazard to non-human biota, public health, and agriculture [49,50,51,52,53]. Therefore, understanding the mechanisms of the soil bacterial communities’ response to ionizing radiation is crucial for identifying issues in soil functioning and making informed decisions regarding the reclamation of radioactively contaminated territories.

The culture-dependent approach traditionally employed in microbiological research is notably time-consuming, involving the isolation and cultivation of bacteria on specific culture media. While this method enables the comprehensive exploration of the taxonomic and functional characteristics of isolated organisms, it exhibits a limited efficiency in community analysis, typically identifying only around 5% of the dominant representatives within a given community [54,55,56]. In contrast, the utilization of modern next-generation sequencing (NGS) techniques facilitates the acquisition of detailed insights into the taxonomic structure and functional profile of entire bacterial communities. By combining a metagenomic strategy that entails the extraction of total DNA from the soil with a subsequent NGS analysis, researchers gain access to a profound examination of the vast nonculturable diversity that was previously inaccessible [57]. Metagenomic approaches have emerged as primary methodologies for investigating the diversity of soil communities, supplanting the limitations of culture-dependent methods [56,57,58,59]. These advanced techniques offer a more comprehensive and efficient means to study microbial communities in soil environments.

There are numerous publications that explore bacterial communities in radioactive contaminated soils or those irradiated under laboratory conditions using both culture-dependent and molecular–genetic approaches. These study results are interesting to discern common changes in the community’s taxonomic or functional transformation and adaptation mechanisms.

The level of radioactive contamination is typically not extreme for soil bacterial communities, primarily due to their high radiotolerance even in the presence of high radionuclide activity concentrations. However, the unique nature of environmental contamination exerts long-term effects through multifactorial exposure. Consequently, uncertainties arise that may impede the analysis of community structure and composition data under varying chemical, physical, and biological influences obtained in different studies. In addition to ionizing radiation, various factors, such as the soil moisture, pH levels, organic carbon content and the availability of macro- and micro-elements, the chemical toxicity of radionuclides, and the presence of other toxicants (like heavy metals and oil products), influence the activity of soil organisms. Identifying trends within extensive datasets can enhance the efficacy of selecting and estimating radionuclides and external ionizing radiation exposure amidst other environmental influences.

The objective of this review was to summarize the existing scientific data on alterations in the taxonomic and functional compositions as well as the adaptations of soil bacterial communities in highly radioactive environments with a focus on identifying the primary factors influencing these alterations. Furthermore, an endeavor was made to determine whether the ionizing radiation or chemical toxicity of radioactive elements plays a pivotal role in influencing the structure and diversity of soil bacterial communities. This review considered studies in which the radioactive contamination of the soil was estimated using measurements of the radioactive background, pollution density, radionuclide activity concentrations, or concentrations in the soil. Additionally, studies in which irradiation doses were calculated were included. The review covered the findings derived from the analysis of 55 articles spanning the period from 1996 to 2024, sourced from databases such as PubMed (https://www.ncbi.nlm.nih.gov/ (accessed on 28 March 2024)), Google Scholar (https://scholar.google.com/ (accessed on 28 March 2024)), and eLibrary (https://www.elibrary.ru/ (accessed on 28 March 2024)). The selection of articles was based on specific keywords, including ‘bacterial communities’, ‘radioactive contamination’, ‘ionizing radiation’, and ‘taxonomic diversity’. This methodological approach underpinned the structure and focus of our review, providing a comprehensive examination of the subject matter.

One of the main challenges in summarizing the data from these studies was the lack of standardization in their methodologies. There were significant differences in the structure of the analyzed studies, the methodological approaches used to estimate the state of bacterial communities, and the parameters chosen to characterize the radioactive situation.

## 2. Taxonomic Diversity and Structure of Soil Communities after Radiation Exposure

Table 1 provides a summary of studies conducted from 1996 to 2024, focusing on the impact of radioactive soil contamination and external gamma irradiation on bacterial diversity [15,19,20,25,26,27,29,30,34,35,60,61,62,63,64,65,66,67,68,69,70,71,72,73,74,75,76,77,78,79,80,81,82,83,84,85,86,87,88,89,90,91,92,93,94,95,96,97,98,99,100,101,102,103]. Particular studies are discussed below in the correspondent sections.

### 2.1. Soils Contaminated by Artificial Radionuclides

Nuclear tests, the operations of nuclear facilities, and accidents stand out as the primary sources of artificially derived radionuclides released into the environment [49,104,105]. Notably, events such as the Chernobyl and Fukushima nuclear power plant accidents led to the extensive contamination of large areas with radionuclides like ^137^Cs, ^90^Sr, and others [106,107,108,109]. Nuclear tests conducted by countries such as the USSR, USA, and France have resulted in the pollution of vast territories with ^137^Cs, ^90^Sr, and isotopes of uranium and plutonium [110,111,112]. The persistent soil radioactive contamination by long-lived radionuclides continues to pose a significant environmental hazard to ecosystems and non-human species to this day [113].

The initial studies on bacterial communities in soils contaminated with radionuclides primarily focused on the examination of bacteria capable of growing solely on cultural media. Thus studies, as indicated in Table 1 [30,60,61,62,64,114], revealed a predominance of aerobic species within these communities. However, during the early years following the Chernobyl accident, when the radioactive background levels reached 629 kBq/kg [60,61], a decrease in the diversity of chemoorganotrophic bacteria and a reduction in the abundance of cellulose-degrading, nitrifying, and sulfate-reducing bacteria were noted in the soil. Furthermore, the authors observed a significant presence of *Bacillus cereus* from the Bacciliota phylum [61] and representatives of the *Methylobacterium* genus from the Pseudomonadota phylum [60,61]. Additionally, other studies employing a culture-dependent approach reported no differences in the diversity and ability of bacteria in contaminated soils [62,64]. In fact, an increase in ^137^Cs activity concentrations in the soil up to 1700 kBq/kg [62] resulted in the enhanced diversity of cultivated bacteria compared to the reference area. However, the authors attributed these results not to the radioactive contamination level but to the high organic material content of the soil. The physical and chemical characteristics of the soil, rather than the ^137^Cs activity concentration, were the primary factors influencing the diversity and abundance of cultivated bacteria in the study [64].

Recent studies performed with NGS have revealed findings regarding the alpha diversity of bacterial communities in radioactively contaminated areas when compared to uncontaminated regions. The research by [65,66] demonstrated an increase in alpha diversity within bacterial communities in the contaminated zone, likely attributed to the relatively low activity concentration of ^137^Cs in the soil, which was insufficient to induce significant effects on bacteria (Table 1). Conversely, studies by [27,29] reported no alterations in bacterial diversity, with [27] noting a slight decrease specifically in areas with elevated activity concentrations of ^137^Cs and ^134^Cs exceeding 500 and 200 kBq/kg, respectively. Notably, [26] observed a significant reduction in the alpha diversity of bacterial communities in cases of more complex contamination, involving high activity concentrations of not only ^137^Cs, but of ^90^Sr, ^241^Am, and ^154^Eu in the soils of the Chernobyl exclusion zone. The use of culture-dependent methodologies has also provided insights into the effects of ^137^Cs contamination on bacterial communities following nuclear tests. Gu and colleagues [30] reported a decrease in bacterial diversity when high levels of ^137^Cs contamination were present, but in cases of moderate contamination, the alpha diversity was slightly increased. This tolerance of bacterial communities can be attributed to changes in their taxonomic structure and the dominance of species resistant to ionizing radiation, such as *Geodermatophilus bullaregiensis* [27] or representatives from the Prolixibacteraceae (Bacteroidota) and Methylococcaceae (Pseudomonadota) families [29].

The increase in the irradiation dosage from artificial radionuclides has led to a reduction in soil bacterial diversity and alterations in their taxonomic composition. The prevalence of specific phyla in contaminated areas is influenced by the soil characteristics, such as pH levels, moisture content, and organic carbon levels, which, in turn, are features of the geographical location of the research site. Predominantly observed phyla include Pseudomonadota, Acidobacteriota, and Actinomycetota in most instances [26,27,30,65]. Verrucomicrobiota representatives in contaminated soils have shown a tendency to decrease [26,65,87], while Chloroflexota have exhibited an increase [27,65].

Distinct operational taxonomic units (OTUs), identified down to the genus/species level, that dominate in contaminated areas are commonly associated with adverse environmental conditions [27,29,65,114]. These conditions include cold temperatures, low oxygen levels, acidity, alkalinity, salinity, and the presence of metals in soils. Examples of such OTUs include *Geodermatophyllus bulgariensis* [27,115], *Truepera radiovictrix* [27,116], and *Rubrobacter taiwanensis* [27].

In the Okuma area, which stands as one of the most heavily contaminated regions following the Fukushima accident, various microorganisms thriving in harsh environments have been identified [27]. These include the piezotolerant bacterium *Marinilactibacillus piezotolerans* [117], the volcanic inhabitant *Rubrobacter spartanus* [118], the nitrogen-fixing species *Pleomorphomonas koreensis* [119], and the selenate-respiring bacterium *Thauera selenatis* [120] (all species are mentioned in [27]).

### 2.2. Soils with Enhanced Levels of Naturally Occurring Radionuclides and Heavy Metals

The landscape and geochemical features of territories, mining activity (uranium, coal, phosphates, and rare-earth elements), oil extraction, and many other factors affect the concentrations of naturally occurring radionuclides (^238^U, ^226^Ra, ^232^Th, ^40^K, etc.) and associated heavy metals and metalloids in soils ([38,121,122,123,124,125] and others (Table 1)). The conditions of the increased background of natural radioactivity solely caused by the geochemical features of the underlying rock may be typical for some soil bacterial communities, but their structure may differ from that in reference territories. However, in cases when the increase in the natural radiation background is the result of anthropogenic activity, bacterial communities are forced to adapt to new unfavorable conditions, which leads to a change in their taxonomic structure and community composition.

Most available studies have focused on the state of microbial communities under the influence of elevated concentrations of naturally occurring radionuclides, which have entered the soil due to uranium mining and the disposal of uranium industry waste [19,67,68,69,70,71,72,74,75,76,77,79,81,84]. Much less research has been dedicated to the effect of exclusively natural soil radioactivity on bacterial communities [15,73,86,88], the consequences of nuclear industry plant [85,87] and oil extraction activities [35,78], as well as the results of laboratory experiments with the introduction of radioactive substances or materials containing uranium isotopes into the soil [90].

The research results on bacterial communities from areas with elevated levels of naturally occurring radionuclides and HMs, regardless of climatic conditions, soil pH, and the nitrogen, phosphorus, and organic carbon contents, predominantly show a significant decrease in alpha diversity compared to the nearby reference areas (Table 1). The bacterial diversity decreased less in the area with mostly RN contamination than in the neighboring areas contaminated with RN and HMs or only HMs (for example, [126,127]). However, there are exceptions. Thus, in studies of the bacterial communities of soils technogenically contaminated with U and HMs [84] and soils with increased natural radioactivity (^238^U, ^232^Th, and ^40^K) [15], no statistically significant changes in diversity were found in areas with different RN or HM concentrations in the soil. Significant changes in the taxonomic composition of bacterial communities at the phylum level are usually not recorded; however, statistically significant rearrangements at the level of lower taxa that are already adapted to live in an RN environment are common ([15,20,80,82,83] and others from Table 1).

Frequently, bacteria capable of changing the degree of the oxidation of elements dominate the composition of communities in contaminated territories (Appendix A), for example, iron-reducing [73,80], capable of U(VI) reduction [73,80], and sulfate-reducing bacteria [80,128]. In a contaminated environment, bacteria that produce [82] and are resistant to antibiotics [15,79] and metals [15,79] thrive, as well as some halophytes [15,79]. As a result, in territories with elevated levels of naturally occurring radionuclides, *Acidithiobacillus ferrooxidans* and several Pseudomonas species have an advantage [67,68,70,83,84,86]. The bacterium *A. ferrooxidans* is known for its ability to inhabit extremely acidic conditions and oxidize Fe^2+^ [129], while representatives of the genus Pseudomonas are involved in the redox transformations of U(VI) and exhibit resistance to uranium and HMs [70,130,131].

### 2.3. Laboratory Experiments with External Irradiation of Soils

Of the studies investigating the influence of radioactivity on bacterial diversity, some inquiries delve into the repercussions of external gamma radiation. A direct comparison between laboratory findings and those derived from environmental settings, characterized by markedly lower irradiation doses, proves challenging. Nonetheless, insights gleaned from scenarios involving exceptionally high external irradiation doses highlight the mechanisms that allow bacteria to thrive in adverse conditions and try to separate them.

Research examining the impact of γ-radiation on the soil bacterial diversity and composition under laboratory conditions is predominantly centered on identifying radiotolerant species and genera that could be used for the bioremediation of radioactively contaminated sites. These studies range from doses as low as 5 Gy [92], investigating arbuscular mycorrhiza in *Holcus lanatus*, to doses as high as 100 kGy [100], simulating the Martian environment.

The taxonomic structure and composition of bacterial communities often change, and their diversity significantly decreases when soils are exposed to high doses of radiation [96,99,101,103]. In such cases, the eliminated species are frequently replaced by more tolerant bacteria, fungi, or microalgae [99,101]. However, in certain instances, such as hydrocarbon-polluted soils [96] or acid brown earth soils [95], higher doses up to 10 kGy did not result in a decrease in the soil bacterial diversity or even cause an increase in the case of irradiation up to 3 kGy of garden clayey soil [96].

The prevalence of Deinococcota [93,94,98,101,103] in desert bacterial communities exposed to high levels of external irradiation, often accompanied by a significant reduction in other taxonomic groups [102], represents a key characteristic of their taxonomic composition. Members of the Deinococcota phylum exhibit remarkable radiotolerance, with notable examples including the exceptionally resilient *Deinococcus radiodurans*, capable of surviving exposures exceeding 5 kGy [132]; *Deinococcus ficus*, which can withstand 3 kGy (equivalent to an absorbed dose of 18 kGy) [102]; and *Deinococcus guangxiensis*, exhibiting a 10% survival rate at 9.8 kGy [133].

Several representatives of the Bacteroidota phylum have been identified as exceptionally radioresistant bacteria, such as the genus *Hymenobacter* [102]. Within this genus, certain species exhibit remarkable tolerance to high levels of irradiation: *H. taeanensis* can withstand doses up to 3 kGy, *H. swuensis* up to 7.3 kGy, and *H. xinjiangensis* up to 8 kGy [134,135,136]. Inhabitants of the Taklimakan Desert [98] have been found to include the radioresistant [137] genera *Rufibacter* and antibiotic-resistant *Pontibacter* [138].

An increased number of certain members of the Pseudomonadota phylum have been observed in soils subjected to laboratory irradiation conditions [94,98]. This includes thermophilic and radioresistant *Microvirga* [98,139] and microaerophilic, antibiotic-resistant *Lysobacter* [98,140,141]. Within the Chloroflexota phylum, which includes hyperthermophilic species [142], there have been documented increases in certain cases [96,101]. Similarly, the Bacillota phylum [98,103] has shown an increase, encompassing thermophilic and halophilic anaerobic species such as *Anaerosporobacter* [103,143].

## 3. Mechanisms of Resistance and Adaptation in Bacterial Communities in Radioactively Contaminated Soil

In conditions of radioactive soil contamination or exposure to high doses of γ-radiation in a laboratory experiment, changes occur in the structure of ecologically trophic groups of soil microbial communities due to habitat modifications. The shift in the functional composition of the soil microorganisms reflects changes in the taxonomic profile of the community, the chemical composition of the soil, and the availability of essential and toxic elements. The change in the taxonomic structure of soil bacterial communities under radiation exposure may result in the change in the distribution of genes involved in different biological processes. Consequently, this triggers the restructuring of the existing metabolic pathways and the activation of new ones, ensuring the stable existence of the community in the environment contaminated by radionuclides [20,25,26,102]. It is expected that, in the structure of bacterial communities in radioactively contaminated soils, as a rule, bacteria with a greater resistance to radiation compared to other groups, as well as with a wide range of other properties, which allow them to exist in these unfavorable conditions, will gain an advantage or begin to dominate. For example, multiple-stress tolerance has been demonstrated for five radioresistant bacterial isolates obtained from various habitats [144].

In Appendix A, a list of the genera and species of bacteria that the authors of the corresponding studies proposed as dominant in soils from radioactively contaminated sites or irradiated in the laboratory is presented. Usually, the species or genera mentioned as dominant are not characterized as radiotolerant. The main components of the contaminated sites discussed in this review are metals and radionuclides. Based on the mechanism proposed in [145,146,147,148] to provide bacterial tolerance to ionizing radiation and toxic compounds, a list of properties, including metal and antibiotic resistance, high DNA repair efficiency, resistance to low O_2_, the presence of antioxidative enzymes, thermophilicity, halotolerance, involvement in redox reactions, and others, was chosen. Based on this list ([11,70,98,115,116,117,118,119,120,128,130,131,134,137,138,139,140,141,143,149,150,151,152,153,154,155,156,157,158,159,160,161,162,163,164,165,166,167,168,169,170,171,172,173,174,175,176,177,178,179,180,181,182,183,184,185,186,187,188,189,190,191,192,193,194,195,196,197,198,199,200,201,202,203,204,205,206,207,208,209,210,211,212,213,214,215,216,217,218,219,220,221,222,223,224,225,226,227,228,229,230,231,232,233,234,235,236,237,238,239,240,241,242,243,244,245,246,247,248,249,250,251,252,253,254,255,256,257,258,259,260,261,262,263,264,265,266,267,268,269,270,271,272,273,274,275,276,277,278,279,280,281,282,283,284,285,286,287,288,289,290,291,292,293,294,295,296,297,298,299,300,301,302,303,304,305,306,307,308,309,310,311,312,313,314,315,316,317,318,319,320,321,322,323,324] from Appendix A) and the current literature available, an attempt was made to analyze the possible mechanisms that cause the beneficial properties of these bacteria in soils with an increased content of artificial and naturally occurring radionuclides. It should be noted that bacteria possessing one of the properties listed above can simultaneously be resistant to other environmental factors. This is due to the diversity of environmental factors in any natural habitat, especially in territories with extreme conditions.

In this section, we focused our attention on examining and discussing the functional characteristics of bacterial communities, primarily in radioactively contaminated soils, as the mechanisms of soil bacterial communities functioning under γ-ray exposure in laboratory studies are better examined and more evident. It should be noted that the survival of bacteria after exposure to high doses is possible due to the combination of the properties of the organisms ensuring the integrity of the genetic information, as proven in the studies of the extremely high radioresistance of *D. radiodurans* and other bacteria [147,325].

### 3.1. Resistance to Heavy Metals and Antibiotics

The tolerance of soil bacteria to toxic heavy metals and antibiotics shares several common mechanisms, including an ability to avoid the uptake of toxic compounds and to rapidly remove or isolate the absorbed molecules [145,146,326]. Furthermore, heavy metal contamination can contribute to the dissemination of antibiotic resistance through collaborative and indirect selection processes [85]. Typically, natural soil bacterial communities include species resistant to metals and/or antibiotics, such as representatives from the *Staphylococcus*, *Acinetobacter*, *Pseudomonas*, and *Serratia* genera (Appendix A). These resistance genes are localized in both chromosomes (more specific) and in plasmid DNA (less specific, but capable of rapid horizontal gene transfer) [145,146,326].

Among the primary mechanisms of resistance to metals are accumulation in the cell wall, active transportation out of the cell, and intra/extracellular entrapment [145,327]. Additionally, certain bacteria have the capability to alter the oxidation state of elements, particularly transition metals like Fe, Mn, Cr, and others [145,327]. This ability is commonly found in bacteria that utilize redox reactions as an energy source [328], and the presence of such species can benefit the entire community by reducing the bioavailability of toxic elements, including radioactive ones like uranium [130,329,330,331]. Studies have shown an increase in the diversity of Fe-reducing bacterial OTUs in soils contaminated with both radioactivity and metals [76]. The analysis of various studies has revealed a prevalence of dominant chemotrophic bacteria in areas contaminated with artificial radionuclides [114], and more often in regions affected by naturally occurring radionuclides and metals [67,68,73,75,80,84,86].

The mechanisms of antibiotic resistance exhibit a greater diversity compared to resistance to toxic metals. Common strategies shared with various toxicants include the selective penetration of the cell wall and the active transportation of the toxicant out of the cell [146]. Specific to antibiotics, other mechanisms target protein structures, such as replacing sensitive targets with insensitive ones, inducing changes in proteins through point mutations or indels in coding genes, and increasing target production via enhanced expression or gene duplication [146].

The extensive diversity in antibiotic resistance mechanisms is a result of the prolonged co-evolution between antibiotic producers and their targets [332,333]. In recent decades, the hidden diversity stemming from this co-evolutionary arms race has gained momentum due to human antibiotic use [333]. The broader range of targets with specificities has led to a variety of antibiotic resistance mechanisms that are likely more sophisticated than those for heavy metal resistance, which, although numerous in the environment, are comparatively limited.

It is noteworthy that the prevalence of antibiotic-resistant and heavy metal-resistant bacterial groups appears to be comparable among species and genera that dominate in areas contaminated with both artificial and naturally occurring radionuclides. Bacteria such as *Acinetobacter*, *Burkholderia*, *Streptomyces*, *Bacillus*, and *Staphylococcus*, known for their antibiotic resistance [146,149,178,179], have been identified in soils contaminated with artificial radionuclides (refer to Appendix A). These findings include only data on the cultivated species relevant for potential industrial applications involving the production of organic compounds and the development of remediation strategies for contaminated areas, with implications for public health and agriculture.

Conducting comprehensive studies to assess resistance to all metals and known antibiotics across all cultivated bacterial species is impractical. Moreover, the diversity of uncultivated bacteria significantly expands the pool of species capable of growing on artificial media. Therefore, the data presented in Appendix A material are incomplete, suggesting that the actual number of bacteria resistant to metals and antibiotics in radioactively contaminated environments may be higher. Oxidative stress has been linked to an increased frequency of antibiotic resistance genes in soil bacterial communities (for a comprehensive review, see [334]). Studies have reported heightened abundance and diversity of heavy metal and antibiotic resistance genes in regions affected by past nuclear weapon production activities [85]. Similar trends have been observed, with a simultaneous rise in the frequency of membrane transporter genes in soils contaminated with uranium [20,74].

### 3.2. Halotolerance 

The ability to survive at high salt concentrations is due to the resistance to pronounced osmotic and oxidative stresses through ion homeostasis maintenance, the accumulation of osmolytes, and production of enzymes with stable activity in high-salt environments and thermotolerance [335,336]. But the strategy depends on the salt concentration [336]: Species, adapted to extremely high salt, accumulate inorganic ions intracellularly to balance the salt concentration in their environment. Low-salt or fluctuating salinity environments require inert compatible organic solute (osmolyte) accumulation to protect proteins from denaturation and the activation of proton pump and cation K^+^ and Na^+^ transport. Among the bacteria with the ability for halotolerance, *Methylobacterium*, *Flavobacterium*, *Bacillus*, *Marinilactibacillus*, *Halomonas*, *Salinimicrobium*, and others were found in radioactively contaminated soils (Appendix A). In studies in which these bacteria have been identified, the habitat is not characterized by strong salinity, as in salt marshes. The dominant position of these bacteria in conditions of radioactive contamination can be explained by their high resistance to oxidative stress, accompanying halotolerance [336,337]. In some cases, it can be assumed that bacteria have entered the soil of radioactively contaminated areas from external sources, such as the deep-sea representative *Marinilactibacillus piezotolerans* [27], which could have been introduced as a result of a tsunami following an earthquake and causing the accident at the Fukushima Daiichi Nuclear Power Plant.

### 3.3. Redox Activity and Antioxidant Defense Systems

The ancestors of modern chemotrophic organisms were anaerobic. The evolutionary emergence of a system of antioxidant enzymes, such as catalase, superoxide dismutase, oxidase, and others, predates the presence of free oxygen in the Earth’s atmosphere [338]. This system laid the foundation for the evolution of mechanisms conferring tolerance to oxygen. It is known that most antioxidant enzymes in bacteria are encoded by single genes that make them suitable for horizontal gene transfers and duplications [339]. This feature provides, in short, from an evolutionary perspective, time to increase the fitness of an organism. Catalase- and oxidase-positive strains were isolated from desert soils exposed to 3 kGy [98]. In addition, many bacteria that dominate radioactively contaminated sites have such activity (Appendix A). Also, the presence of catalase and/or oxidase activity may be associated with the ability to utilize redox reactions as an energy source, as seen in bacteria of the *Aquabacterium*, *Halomonas*, and *Pseudomonas* genera, or resistance to heavy metals and antibiotics, as observed in Acinetobacter, Staphylococcus, Ochrobactrum, Pseudomonas, Pseudolabrys, and Robiginitalea.

In addition to the enzymatic detoxification of reactive oxygen species, the ability to produce low-molecular-weight compounds with radioprotective and antioxidant properties, such as carotenoids, apparently plays a role in the formation of resistance to the effects of ionizing radiation. The majority of bacteria dominant in communities of radioactively contaminated areas (Appendix A) are characterized by colonies colored yellow, orange, red, or pink, which may indicate the presence of carotenoids [340]—an important component of the antioxidant defense of prokaryotic cells [341]. Pigmented bacteria are common in extreme habitats, ranging from the cold deserts of Arctic and Antarctic regions to arid regions and saline lakes [98,340]. The antioxidant properties of carotenoids, providing protection from UV radiation, contribute to the survival of bacteria in other unfavorable conditions. Carotenoids have been found in bacterial isolates obtained from soils from the high background radiation area (HBRA) of the Chavara-Neendakara placer deposits (Kerala, India) [88] and the radioactive site of Misasa (Tottori, Japan) [340,341]. Also, pink-colored colonies of bacteria of the genus *Methylobacterium* were isolated from soils contaminated with radionuclides as a result of the Chernobyl accident [60]. The proportion of pigmented bacteria, including those with carotenoid presence, in the Bacillaceae family in the bottom sediments was significantly higher in samples from regions with a high radiation background, as a result of the Chernobyl nuclear power plant (ChNPP) accident, compared to areas not contaminated with radionuclides. Additionally, the increase in radiation background levels led to a significant growth (by 69.7%) in the proportion of bacilli capable of synthesizing multiple types of pigments [342]. Conversely, the loss of the ability to synthesize carotenoids due to mutation significantly reduced the radioresistance of *Deinococcus radiodurans*, as noted by [343].

### 3.4. Defense Mechanisms Associated with Thermophilicity

It is known that thermophilic bacteria can exhibit increased resistance to ionizing radiation. The evolution of prokaryotes began in the early Earth, when natural radioactivity, including the radiation of the early Sun and heavy elements, ^232^Th, ^235^U, and ^238^U, and ^40^K, was an intrinsic component of the natural environment [325]. It is not coincidental that resistance to ionizing radiation correlates with resistance to UV radiation and high temperatures, and the most radioresistant among modern bacteria belong to the group of hyperthermophilic organisms.

The majority of the radioactively contaminated sites studied are not associated with arid territories, where the soil heats up to high temperatures and the natural bacterial communities exhibit a wide variety of extremophiles. However, among the cultivated isolates from soil samples collected in the early years after the Chernobyl nuclear power plant accident, radioresistant bacteria of the genus *Methylobacterium* dominated [60], representatives of which have the ability to survive in conditions of high temperatures, UV radiation, high salinity, and drought [159]. Additionally, in the soils around the Fukushima nuclear power plant, the radio- and thermotolerant bacteria [115,116,177] *Geodermatophillus bullgariensis*, *Truepera radiovictrix*, and *Rubrobacter taiwanensis* [27] (Appendix A) were registered among the dominant species. Thermophilic bacteria from the genera *Alicyclobacillus* and *Bellilinea* were found among the dominant species in areas with increased natural radioactivity [20,80].

Survival in such an inhospitable environment requires the presence of excellent antioxidant protection [147,148,344,345] and highly effective DNA repair mechanisms. An effective strategy to minimize oxidative damage for many bacteria is to thrive in environments with reduced oxygen levels. Interestingly, microaerophilic organisms are commonly found among both the most sensitive and the most resistant to ionizing radiation bacteria [148]. For example, among those found in radioactively contaminated soils, *Gallionella* and *Flavobacterium* bacteria can exist in microaerophilic conditions (Appendix A). The acceleration of DNA damage repair is typical for a number of polyploid/merooligoploid bacteria common among Pseudonadota [346], like *Methylobacterium* and *Pseudomonas*, as shown in Appendix A. In addition to enhanced antioxidant system activity, it can be achieved by increased nucleoid condensation [147] and the ability to assemble intact chromosomes from fragmented pieces. This latter ability is known for some hyperthermophilic bacteria, including radioresistant ones [147,346].

So, the proposed adaptation scheme (Figure 1) for the bacterial community in response to radioactive soil contamination is founded on comparing lists of dominant species (as presented in Appendix A) with a compilation of the features that facilitate bacterial survival in contaminated environments. It has been observed that, while certain bacteria dominating contaminated areas may not be explicitly classified as radiotolerant, the presence of specific features enables their survival in the presence of metals and radionuclides. In radioactively contaminated communities, almost all dominated bacteria present had antioxidant enzymes (such as catalase, oxidase, or superoxide dismutase) or low-molecular-weight antioxidants (96%). Furthermore, a significant majority displayed resistance to metals and antibiotics (96%), alongside possessing protective redox systems (68%). Additionally, a substantial portion of the dominant bacterial species demonstrated halotolerance (72%), while a notable percentage were thermophiles with heightened DNA repair mechanisms (44%).

## 4. Ionizing Radiation in the Forming of Soil Bacterial Communities

The review of the studies examining the impact of external irradiation exposure on soil revealed the remarkable radiotolerance of bacterial communities. The doses that induce significant changes in the taxonomic composition of soil bacterial communities at the phylum level are practically unattainable in landscapes affected by artificial or naturally occurring radionuclides. Adverse effects primarily manifest in the restructuring of the taxonomic structure and a decrease in diversity at high exposure levels. Consequently, the toxic exposure to radionuclides emerges as a key factor driving changes in community composition. The high prevalence of bacteria resistant to metals, antibiotics, and polycyclic aromatic hydrocarbons in radioactively contaminated soils support this hypothesis.

On the other hand, chronic exposure and the interaction of high-linear energy transfer radiation sources, such as uranium isotopes and certain decay products like ^241^Am, with biological macromolecules in naturally contaminated environments highlight the significance of ionizing radiation from radionuclides present in contaminated soils. Additionally, within irradiated communities, bacteria with highly efficient antioxidant systems and the ability to endure extreme conditions, leading to genome stabilization, have been identified. Both adaptation mechanisms can prove advantageous in the face of genotoxic and pro-oxidant exposures.

The physical and chemical characteristics of the soil, such as moisture, pH levels, granulometric composition, temperature, oxygen levels, organic matter content, NPK levels, dominant vegetation, among others, play a pivotal role in shaping the adaptation of bacterial communities in radioactively contaminated soils [27,62,65,85,87]. Furthermore, the selection process is significantly influenced by the spectrum of radionuclides present and the spatial heterogeneity of contamination [18,112,347]. Assessing the contribution of ionizing radiation to community-based biological effects is only possible under strictly controlled laboratory conditions. It is essential to note, however, that the response of organisms in natural and laboratory settings may differ significantly [348]. Despite these differences, some patterns in the taxonomic diversity of bacterial communities from radioactively contaminated areas have been observed, and certain structural differences between contaminated and reference communities have been distinguished for various radioecological situations.

To assess the long-term effects of radiation contamination, it would be appropriate to identify the functional potential of the genera/species of bacteria that thrive in the transformed community inhabiting the contaminated soil. This analysis could help to elucidate the causes of such changes and provide valuable insights into the resilience and adaptability of microbial communities in the face of environmental stressors. It would indeed be important to consider the main physical and chemical characteristics of the soil and their changes as a result of technogenic influence when assessing the long-term effects of radiation contamination. Additionally, understanding the influence of such changes on the formation of soil bacterial communities is crucial. To gain an in-depth understanding of these relationships, laboratory experiments on determining the influence of changes in the basic characteristics of the soil on the structure of communities against the background of the same level of radiation pollution would be highly beneficial.

When studying the communities inhabiting radioactively contaminated soils, it would be beneficial to focus on analyzing the whole complex of metabolic pathways represented in the community. This can be achieved by examining the frequency of occurrence of genes that confer resistance to contamination, which becomes more feasible with the enrichment of databases of annotated bacterial genomes and the enrichment of metataxonomic research data through the additional sequencing of amplicons of stability genes propagated by horizontal transfer or through the metagenomic analysis of genome-wide libraries.

## 5. Conclusions

Radioactive contamination has the potential to alter the taxonomic structure of soil bacterial communities, favoring tolerant representatives while decreasing the number of or eliminating sensitive ones. Exposure to external ionizing radiation in laboratory conditions has been shown to result in a significant decrease in diversity, with the dominance of separate highly radiotolerant taxonomic groups. The radioactive contamination of the soil is more likely to lead to a decrease in the alpha diversity and rearrangements in the taxonomic structure. In addition to radioresistance, bacteria that dominate in contaminated environments exhibit tolerance to metals and antibiotics, as well as the ability to produce low-molecular-weight antioxidants, halotolerance, and thermophilic characteristics, with an increased ability to maintain the stability of macromolecules, including DNA.

In studies of the radioactive contamination of the soil, it is challenging to quantify the differential contribution of radioactive exposure and the chemical toxicity of radionuclides. Additionally, the bacterial community transformation following the introduction of radionuclides into the soil is influenced by the initial community composition and the physical and chemical features of the soil.

Thus, the systematization of the data presented in the literature can make a significant contribution to understanding the processes occurring in the bacterial communities of radioactively contaminated soils, as well as the main factors affecting the diversity and composition of prokaryotic soil communities during the radioactive pollution of their habitat.

## Figures and Tables

**Figure 1 microorganisms-12-00733-f001:**
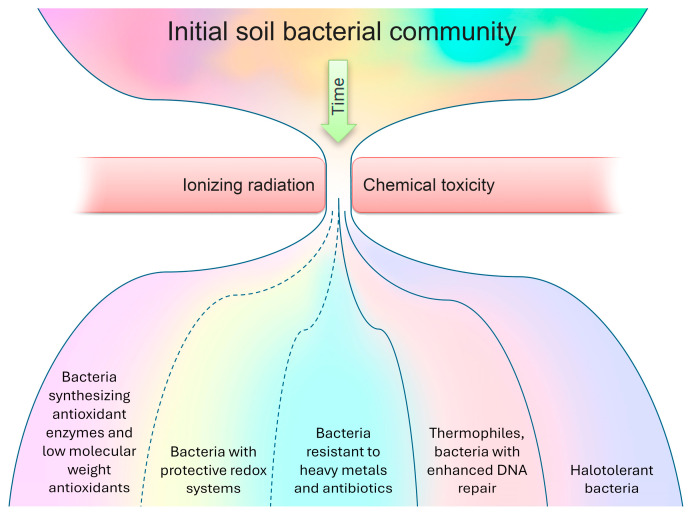
The proposed adaptation scheme for a bacterial community in response to radioactive soil contamination.

**Table 1 microorganisms-12-00733-t001:** Effects of the radioactive contamination of the soil and external gamma irradiation on the taxonomic diversity of bacterial communities.

Radiation Situation	Characteristics of Radiation Factor ^1^	Effect on Taxonomic Diversity	Reference
**Soils Contaminated by Artificial Radionuclides**			
Chernobyl accident	0.037–37,000 kBq/kg (10^−3^–10^−9^ Ci/kg)	↓	[60]
Chernobyl accident	0.074–629 kBq/kg (0.002–17.0 µCi/kg)	↓	[61]
Chernobyl accident	^137^Cs: 61–750 kBq/kg (61–750 Bq/g)	↑↓	[62]
Chernobyl accident	^137^Cs: 3.33–5395.5 mBq/m^2^	↓ Rhodococcus, Pseudomonas, and Aeromonas species; ↑ Mycobacteria	[34]
Chernobyl-like contamination	^137^Cs: 14,700–15,900 kBq/kg ^90^Sr: 10,347–11,167 kBq/kg	↓	[63]
Nuclear tests (northwestern China)	^137^Cs: 100–10,000 Bq/kg	↓ The number of isolates; ↓ Richness in the most contaminated site	[30]
Underground nuclear explosion Kraton-3 (Sakha Republic, Russia)	0.08–1.7 µSv/h (8–170 µR/h) ^137^Cs: 0–29,903 Bq/kg	↑↓	[64]
Chernobyl accident	^137^Cs: 0.35–750 kBq/kg	↑	[65]
Chernobyl accident	^137^Cs: 4.21–1680 kBq/kg ^90^Sr: 1.9–209.1 kBq/kg ^154^Eu: 0.048–3600 kBq/kg ^241^Am: 0.06–35 kBq/kg	↓	[26]
Fukushima accident	^137^Cs: 0.4–34.43 kBq/kg		
Fukushima accident	^134^Cs: 1–207,000 Bq/kg ^137^Cs: 10–563,000 Bq/kg	↓ At the most contaminated site	[27]
Nuclear tests (Uyghur Autonomous Region, China)	^137^Cs: from 10 to >60 Bq/kg	↑	[66]
Chernobyl accident	0.04–130 µSv/h	↑↓	[29]
**Soils with elevated levels of naturally occurring radionuclides and heavy metals**			
Uranium mining (Dresden, Germany)	U: 0.75–3.0 mg/L Th: 0.2–0.4 mg/L	↑↓	[67]
Uranium mining (Dakota, USA)	U: 2–8 mg/kg	↓ At the most contaminated sites	[68]
Uranium mining (Monchique and Urgeiriça, Portugal)	U: 10–22 mg/L	↑↓	[69]
Uranium mining (Jharkhand, India)	U: 1.2 mM; Th: 0.04 mM	It was not compared to the control area	[70]
Uranium mining (Cunha Baixa, Portugal)	Radioactivity: 358–1983 Bq * (358–1983 cps)	↑	[71]
Uranium mining (Jaduguda, India)	U: 1–20 mg/kg Th: 3–17 mg/kg	↑↓	[72]
Natural soils with high uranium content (Limousin, France)	U (total): 27–255,000 mg/kg (27–255,000 ppm) U (soluble): 0.1–81 µM	↑↓	[73]
Uranium mining (Buhovo, Bulgaria)	U: 26.7–374 mg/kg Th: 17.2–22.1 mg/kg	↓	[19]
Uranium mining (Domiasiat, India)	U (soil): from <10 to 1200 mg/kg Uranyl nitrate: 100 µM–2 mM	↑ Diversity of U-tolerant bacteria at medium U concentrations	[74]
Uranium mining (Jharkhand, India)	U: 11–290 mg/kg	↓	[75]
Uranium mining (Ronneburg, Germany)	U: 5–1569 μg/g	Microbial communities have changed, but not because of U	[76]
Uranium mill tailings (southeastern China)	^238^U: 5.2–48.1 mg/kg ^232^Th:12.7–16.9 mg/kg ^226^Ra: 0.06–0.17 Bq/g ^40^K: 1.29–2.27 mg/kg	↓	[77]
Uranium tailings (southern China)	U: 4.31–48.10 mg/kg	↓	[20]
Thorium mining (Sakha Republic, Russia)	0.15–0.42 μSv/h (15–42 μR/h) ^232^Th: 34–1535 Bq/kg	↑↓	[64]
Radioactive oily waste (Tatarstan, Russia)	^226^Ra: 21–2739 Bq/kg ^232^Th: 32–916 Bq/kg ^40^K: 271–311 Bq/kg	↓	[35]
Radioactive oily wastes (Tatarstan, Russia)	^226^Ra: 0.01–1 kBq/kg ^232^Th: 0.021–0.65 kBq/kg	↓	[78]
Uranium mining (Erode, India)	U: 32.4 Bq/kg	↓	[79]
Uranium mining (Ranger Uranium Mine, Australia)	0–4000 mg/kg	↓ At 4000 mg/kg	[80]
Uranium tailings (Beishan, China)	^238^U: 3.42–12.3 mg/kg ^232^Th: 12.6–16.5 mg/kg	↓	[81]
Uranium mining (Ranger Uranium Mine, Australia)	U: from 2 to >900 mg/kg	↑↓	[82]
Uranium mining (Paukkajanvaara, Finland)	^226^Ra: from <0.1 to 21,000 Bq/kg (0.0–21 Bq/g)	↓ At the most contaminated site	[83]
Uranium mining (Sichuan, China)	U: 11.92–54.99 mg/kg	↑	[84]
Activities of the former nuclear weapon production facility (South Carolina, USA)	^238^U: 1.03–6.29 mg/kg	↓	[85]
Natural radiation intensity of radon (Incheon, Korea)	^222^Rn: 920–3367 Bq/m^3^	↓	[86]
Phosphate and nuclear industries (Grote Nete, Belgium)	^226^Ra: to 3750 Bq/kg ^238^U: to 200 Bq/kg ^210^Pb: to 1000 Bq/kg ^241^Am: to 225 Bq/kg ^137^Cs: to 300 Bq/kg ^60^Co: to 12 Bq/kg ^232^Th: to 30 Bq/kg ^228^Th: to 45 Bq/kg ^228^Ra: to 30 Bq/kg	↓ Alpha diversity; Microbial communities exhibited the most distinct composition in regions characterized by the highest concentrations of radionuclides and heavy metals	[87]
Abnormally high background radiation (Kerala, India)	21 μSv/h (21,000 nGy/h)	↓ At the phylum level; ↑ At the genus level	[88]
Increased natural radioactivity (Junggar Basin, China)	^238^U: 27.74–739.13 Bq/kg ^232^Th: 15.95–59.66 Bq/kg ^40^K: 612.5–806.07 Bq/kg	The composition of the microbial community changed in accordance with the gradients of the specific activity of natural radionuclides in the soil	[15]
Uranium mill tailings (Xinjiang, China)	^238^U: 36.52–51.13 Bq/kg ^226^Ra: 40.79–53.18 Bq/kg ^232^Th: 29.09–37.4 Bq/kg ^40^K: 605.89-651.08 Bq/kg	↓	[89]
U tailing dam (Guangdong, China)	U total: 71.12–76.02 mg/kg Uexc: 6.27–7.98 mg/kg Ured: 15.16–33.69 mg/kg Uoxi: 9.61–23.15 mg/kg	↓	[25]
Uranyl nitrate (laboratory contamination, Mianyang, China)	100 mg/kg	↓	[90]
Uranium mining (Qinghai–Tibet Plateau, China)	-	↓	[91]
**External laboratory γ-irradiation**			
^137^Cs γ-irradiation	5–160 Gy	↓ Gram-negative bacteria, especially Pseudomonads; ↑ Gram-positive bacteria	[92]
^60^Co γ-irradiation	0–30 kGy	↓	[93]
^60^Co γ-irradiation	15 kGy	Radioresistant bacterial species have been found	[94]
^60^Co γ-irradiation	0–10 kGy	↑↓	[95]
^60^Co γ-irradiation	1–10 kGy	↑ At low doses; ↓ At higher doses (garden clay soil); ↓ At higher doses (uncultivated clay soil); ↑↓ (Hydrocarbon contaminated soil)	[96]
^137^Cs γ-irradiation	1.8, 4.0 kGy	↓ Radiosensitive bacteria; ↑ Radioresistant bacteria	[97]
^60^Co γ-irradiation	3 kGy	γ-Radiation-resistant bacteria were isolated from the desert samples	[98]
^60^Co γ-irradiation	0–40 kGy	↓	[99]
^60^Co γ-irradiation	100 kGy	High bacterial diversity with changed community structure	[100]
^60^Co γ-irradiation	0.6–18 kGy	↓	[101]
^60^Co γ-irradiation	0.6–18 kGy	↑ In Deinococcota diversity; ↓ Other bacterial diversity	[102]
^60^Co γ-irradiation	0–30 kGy	↓	[103]

Note: ↓—decrease, ↑—increase, or ↑↓—multidirectional changes in diversity. ^1^—a range of values is provided for both background and contaminated sites; *—values were recalculated, assuming that the gamma detection rate is equal to 100%.

## Data Availability

Data are contained within the article and in the Appendix A.

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
