# Peer review of "Taxonomic Diversity and Functional Traits of Soil Bacterial Communities under Radioactive Contamination: A Review"

_microorganisms, 2024, doi:10.3390/microorganisms12040733_

Round 1
Reviewer 1 Report
Comments and Suggestions for Authors
The manuscript represents an interesting overview of articles that consider various aspects of microbial diversity in soils under different modes of exposure to ionising radiation. A great number of articles has been surveyed, and a useful table with dominant species in radioactively contaminated environments has been prepared.
However, before being published, the manuscript requires authors' attention at several significant points.
First of all, the methodology of picking up the articles is unclear. In the end of the introduction, keywords, years, databases used must be explicitly mentioned.
Table 1, the central part of the manuscript, must be carefully checked. All old units (such as Ci) must be converted to Bq to ensure that anyone without radiobiological background can compare data in the table. Old units can follow in brackets if the authors choose to do so. The same for counts per minute (cps). In the text too (i.e., line 132).
In many places in the Table, the superscript is lost (especially columns 1 and 2).
Why in Uranium Mining you use the capital letter in Mining? Other definitions do not have capitalization of words except first one.
Please carefully check English, commas, italics in the column 3 (it doesn’t have lines and therefore it is difficult to specify). For instance, “relatively middle level of radon” is unclear.
The subsections of the table must be highlighted somehow (maybe CAPS LOCK, bold font, or an empty line after each subsection (such as “Soil contaminated by artificial radionuclides))
Column 1, “abnormally high background radiation” or “increased natural radioactivity” probably could benefit from geographical location.
Please specify in the table that you described only external laboratory gamma-irradiation.
The symbol with two arrows (up and down, Line 113) is counterintuitive – I would expect it means multidirectional changes rather than unchanged diversity. Please consider to change the symbol.
Controversial statements
Line 46 – "technologically enhanced natural radioactivity" is defined by U.S. EPA as “Technologically Enhanced Naturally Occurring Radioactive Material (TENORM) ‒ "Naturally occurring radioactive materials that have been concentrated or exposed to the accessible environment as a result of human activities such as manufacturing, mineral extraction, or water processing.”. What do you include then into “radionuclide production for the nuclear industry” (line 47)?
Lines 162-163 – the references cited just above (65, 66, 27, 29) showed no increase, and I suggest to correct the level of certainty here.
Section 2.3 – I believe this is a good moment to explicitly explain why do you consider only external irradiation, since it is almost impossible to compare with any data available from real ecosystems. Also, the general design of those studies is unclear. Is it an artificial bacterial community? Is it real samples from the field? What are the majority of soil samples in these studies and why they have been chosen? What do we know about background microbiomes before irradiation? All this important information is not clear in this part of the review.
Section 3.1 – This is a very important section. However, it can benefit from some improvements. Among them are clear hypothesis about antibiotic resistance in terms of radiation exposure. Currently, the link is unclear. For heavy metals, we can at least talk about RN transport and reallocation and about antioxidant defence system. Why antibiotic resistance is put in the same sentence as “other toxic metals”, Line 324? I can’t see how xenobiotic binding or antiporter activity for antibiotic molecules can help withstand radiation stress. I would like the authors to make this part clearer for readers.
Figure 1 – Since the Figure is based on radioecological research, I believe it is important to reflect all the confounding factors or at least mention them in the legend (especially because there is no property that has been typical for all studies assessed). The percent of studies supporting each branch could be useful (calculated from the Table S1).
Table S1 – Could you consider to include a column with environmental properties? Soil type at least. The Table S1 is still somehow valuable in its current view, but its value (and subsequent citations of the review) can be significantly increased through adding information about edaphic factors.
Minor remarks
Line 43 – please place abbreviation (RN) from the line 50 to here, as this is the first encounter of “radionuclide” term in the text.
Line 52 – term “preparations” seems unclear to me
Lines 70-71 – “NGS and metagenomics” – please, specify. Currently, in most cases, metagenomic studies are done using NGS.
Line 127 – aerobic species
Line 128 – “could grow” – please revise this part, something is missing
Line 129 – a decrease
Line 140 – please consider “physical and chemical”
Line 156 – it is advisable to start sentence with the first author’s surname in this case
Line 169 – please remove “-“
Line 176-181 – double-spacing
Line 193 – have been focused
Line 253 – “elevated levels” – please, reconsider
Line 308 – in terms of what we know about bacterial genomes, I didn’t quite understand the wording “more complex and specific”, could you please explain this?
Lines 380-381 – The sentence is not very clear since obligate anaerobes also can have antioxidant enzymes such as SOD.
Comments on the Quality of English Language
Please read the manuscript again (maybe using Grammarly). English requires your minor attention.
Author Response
Dear Reviewer,
Thank you very much for reviewing our manuscript and making critical analysis and valuable remarks. We have revised the entire manuscript as per your suggestions and tried to take all comments into account. All corrections are indicated with a yellow marker. Answers on the specific comments are lower.
The manuscript represents an interesting overview of articles that consider various aspects of microbial diversity in soils under different modes of exposure to ionising radiation. A great number of articles has been surveyed, and a useful table with dominant species in radioactively contaminated environments has been prepared.However, before being published, the manuscript requires authors' attention at several significant points.
First of all, the methodology of picking up the articles is unclear. In the end of the introduction, keywords, years, databases used must be explicitly mentioned.
Thank you for the remark. The text was added as follows: The review covers the findings derived from the analysis of 55 articles spanning the period from 1996 to 2024, sourced from databases such as PubMed (https://www.ncbi.nlm.nih.gov/), Google Scholar (https://scholar.google.com/), and eLibrary (https://www.elibrary.ru/). The selection of articles was based on specific keywords including 'bacterial communities,' 'radioactive contamination,' 'ionizing radiation,' and 'taxonomic diversity.' This methodological approach underpins the structure and focus of our review, providing a comprehensive examination of the subject matter.
Table 1, the central part of the manuscript, must be carefully checked. All old units (such as Ci) must be converted to Bq to ensure that anyone without radiobiological background can compare data in the table. Old units can follow in brackets if the authors choose to do so. The same for counts per minute (cps). In the text too (i.e., line 132).
Thank you for the remark. In our endeavor to standardize units, we encountered challenges stemming from variations in the data provided. Discrepancies arose from differing measurement methodologies employed by various authors. Some studies quantified uranium concentration in terms of isotopic activity, while others expressed it as the concentration of the chemical element itself. Additionally, absorbed doses and external irradiation doses were reported. To achieve consistency, conversions were made from Ci and cps to Bq, µR/h to µSv/h, ppm to mg/kg, and Bq/g to Bq/kg. Values more than 1000 Bq/kg were transferred to kBq/kg.
In many places in the Table, the superscript is lost (especially columns 1 and 2).
Thank you for the remark. Superscripts were checked and corrected.
Why in Uranium Mining you use the capital letter in Mining? Other definitions do not have capitalization of words except first one.
Thank you for the remark. Capital letters in “Mining” were replaced.
Please carefully check English, commas, italics in the column 3 (it doesn’t have lines and therefore it is difficult to specify). For instance, “relatively middle level of radon” is unclear.
Thank you for the remark. Data in the Table 1 were checked and corrected.
The subsections of the table must be highlighted somehow (maybe CAPS LOCK, bold font, or an empty line after each subsection (such as “Soil contaminated by artificial radionuclides))
Thank you for the remark. The subsection names were highlighted with capital letters and bold font.
Column 1, “abnormally high background radiation” or “increased natural radioactivity” probably could benefit from geographical location.
Thank you for the remark. Locations for increased natural radioactive background areas were added into the table.
Please specify in the table that you described only external laboratory gamma-irradiation.
Thank you for the remark. The external laboratory gamma-irradiation was added to the subsection name.
The symbol with two arrows (up and down, Line 113) is counterintuitive – I would expect it means multidirectional changes rather than unchanged diversity. Please consider to change the symbol.
Thank you for the remark. The footnote was changed for the following: ↓ - decrease, ↑ - increase or ↑↓ - multidirectional changes in diversity, respectively.
Controversial statements
Line 46 – "technologically enhanced natural radioactivity" is defined by U.S. EPA as “Technologically Enhanced Naturally Occurring Radioactive Material (TENORM) ‒ "Naturally occurring radioactive materials that have been concentrated or exposed to the accessible environment as a result of human activities such as manufacturing, mineral extraction, or water processing.”. What do you include then into “radionuclide production for the nuclear industry” (line 47)?
Thank you for the remark. It is a mistake. The sentence is rewritten as follows:
The causes of radioactive contamination can be attributed to several factors: areas with increased natural radioactivity [12-16], technologically enhanced natural radioactivity [17-,18], radionuclide production for the nuclear industry [19-25], radiation accidents [26-29], nuclear tests [30-33], and oil pollution [34-38].
Lines 162-163 – the references cited just above (65, 66, 27, 29) showed no increase, and I suggest to correct the level of certainty here.
Thank you for the remark. These references refer to changes in alpha-diversity as a whole. According to Table 1, in the works [65,66] the diversity increases, [27] - decreases, [29] - changes are multidirectional. It should be noted, however, that with an overall decrease in diversity, there may be a decrease or an increase in the proportion of certain phyla, such as Pseudomonadota, Acidobacteriota, and Actinomycetota in most instances [26,27,30,65]. Verrucomicrobiota representatives in contaminated soils have shown a tendency to decrease [26,65,87], while Chloroflexota have exhibited an increase [27,65]".
Section 2.3 – I believe this is a good moment to explicitly explain why do you consider only external irradiation, since it is almost impossible to compare with any data available from real ecosystems. Also, the general design of those studies is unclear. Is it an artificial bacterial community? Is it real samples from the field? What are the majority of soil samples in these studies and why they have been chosen? What do we know about background microbiomes before irradiation? All this important information is not clear in this part of the review.
Thank you for the remark. The explanation was added to the text:
Within the studies investigating the influence of radioactivity on bacterial diversity, some inquiries delve into the repercussions of external gamma radiation. A direct comparison between laboratory findings and those derived from environmental settings, characterized by markedly lower irradiation doses, proves challenging. Nonetheless, insights gleaned from scenarios involving exceptionally high external irradiation doses highlight mechanisms that allow bacteria to thrive in adverse conditions and try to separate them among others.
Section 3.1 – This is a very important section. However, it can benefit from some improvements. Among them are clear hypothesis about antibiotic resistance in terms of radiation exposure. Currently, the link is unclear. For heavy metals, we can at least talk about RN transport and reallocation and about antioxidant defence system. Why antibiotic resistance is put in the same sentence as “other toxic metals”, Line 324? I can’t see how xenobiotic binding or antiporter activity for antibiotic molecules can help withstand radiation stress. I would like the authors to make this part clearer for readers.
Thank you for the remark. Arguments on the metal and antibiotic resistance were joined in one section because of simultaneous presence of the both in one and the same bacterial strains. Also, similar mechanisms, similar evolution basis of them at least, are used in detoxifying both metals and antibiotics.
We have added this explanation to the text: Heavy metal contamination can help spread sustainability to antibiotics by joint and indirect selection methods [85].
About the line 324 – there was a mistake. It was corrected:
line 323-324 The mechanisms of antibiotic resistance exhibit greater diversity compared to resistance to other toxic metals.
Figure 1 – Since the Figure is based on radioecological research, I believe it is important to reflect all the confounding factors or at least mention them in the legend (especially because there is no property that has been typical for all studies assessed). The percent of studies supporting each branch could be useful (calculated from the Table S1).
Thanks for the remark. The scheme, presented on fig.1 reveals a consistent trend wherein bacteria possessing the specified properties consistently demonstrated an advantage in all radioecological scenarios examined, irrespective of other influencing variables. Among the articles reviewed, only a few species were discussed that exhibited dominance in contaminated areas. Consequently, the lists of dominant species appeared to be arbitrary compilations. Notably, it was unexpected that all these species shared characteristics deemed crucial for defense against radionuclides. These features, underpinned by various mechanisms, enable bacteria to thrive in diverse hostile environments. The percent of studies supporting each branch were added to the text:
So, the proposed adaptation scheme (Figure 1) for the bacterial community in response to radioactive soil contamination is founded on comparing lists of dominant species (as presented in Table S1) with a compilation of features that facilitate bacterial survival in contaminated environments. It has been observed that while certain bacteria dominating contaminated areas may not be explicitly classified as radiotolerant, the presence of specific features enables their survival in the presence of metals and radionuclides. In radioactively contaminated communities, almost all dominated bacteria present have antioxidant enzymes (such as catalase, oxidase, or superoxide dismutase) or low molecular weight antioxidants (96%). Furthermore, a significant majority displayed resistance to metals and antibiotics (96%), alongside possessing protective redox systems (68%). Additionally, a substantial portion of the dominant bacterial species demonstrated halotolerance (72%), while a notable percentage were thermophiles with heightened DNA repair mechanisms (44%).
We also want to keep the diagram in its original form with an indication of the main pollution factors, because in this form it is very clear. We believe that there is no need to add additional factors to the scheme that also affect bacterial communities, because there are a huge number of them and they are very diverse. In addition, the variety of these additional factors is mentioned throughout the text of the review. Displaying them on the diagram seems superfluous.
Table S1 – Could you consider to include a column with environmental properties? Soil type at least. The Table S1 is still somehow valuable in its current view, but its value (and subsequent citations of the review) can be significantly increased through adding information about edaphic factors.
Thank you for the remark. According to your proposal we have added a column “Location (soil type)” to describe main parameters of soil bacteria habitat. Unfortunately, not all authors noted the soil type when describing the studied territory. Thus, we added the location of the contaminated area and when it was possible - the type of the soil.
Minor remarks
Line 43 – please place abbreviation (RN) from the line 50 to here, as this is the first encounter of “radionuclide” term in the text.
Thank you for the remark. We have moved the abbreviation (RN) from line 50 to line 43.
Line 52 – term “preparations” seems unclear to me
Thank you for the remark. The word biological preparations were changed to biostimulants.
Lines 70-71 – “NGS and metagenomics” – please, specify. Currently, in most cases, metagenomic studies are done using NGS.
Thank you for the remark. The text was changed for: MNGS and metagenomic approaches have emerged as primary methodologies for investigating the diversity of soil communities, supplanting the limitations of culture-dependent methods [56-59].
Line 127 – aerobic species
Thank you for the remark. The mistake was corrected: aerobice species
Line 128 – “could grow” – please revise this part, something is missing
Thank you for the remark. The mistake was corrected: The initial studies on bacterial communities in soils contaminated with radionuclides primarily involved the examination of cultivated bacteria that could bacteria could capable of grow solely on cultural media. Thusese studies, as indicated in Table 1 [30,60-62,64,114], determine a relative abundance of aerobice species dominated in communities studied and could grow on cultural media only.
Line 129 – a decrease
Thank you for the remark. The mistake was corrected: However, a reduction in the diversity of chemoorganotrophic bacteria and an decrease in abundance of cellulose-degrading, nitrifying, and sulfate-reducing bacteria were observed in the soils of Chernobyl during the early years following the accident when the radioactive background reached 629 kBq/kg [60,61].
Line 140 – please consider “physical and chemical”
Thank you for the remark. The text “physical-chemical” was changed for “physical and chemical”.
Line 156 – it is advisable to start sentence with the first author’s surname in this case
Thank you for the remark. The text was added: Gu and colleagues [30]...
Line 169 – please remove “-“
Thank you for the remark. “-“ was deleted.
Line 176-181 – double-spacing
Thank you for the remark. The double-spacing has been fixed.
Line 193 – have been focused
Thank you for the remark. It was corrected.
Line 253 – “elevated levels” – please, reconsider
Thank you for the remark. The sentence was changed to the following: Increased number Elevated levels of certain members of the Pseudomonadota phylum have been observed in soils subjected to laboratory irradiation conditions [94,98].
Line 308 – in terms of what we know about bacterial genomes, I didn’t quite understand the wording “more complex and specific”, could you please explain this?
Thank you for the remark. The sentence was clarified and changed for the following: These resistance genes are localized on both the chromosome (more complex and specific) and on plasmid DNA (less specific but capable of rapid horizontal gene transfer) [145,146,326].
Lines 380-381 – The sentence is not very clear since obligate anaerobes also can have antioxidant enzymes such as SOD.
Thank you for the remark. The sentence requires clarification. The text had been changed for the follows:
The ancestors of modern chemotrophic organisms were anaerobic. The evolutionary emergence of a system of antioxidant enzymes such as catalase, superoxide dismutase, oxidase and others predates the presence of free oxygen in the Earth's atmosphere [338]. This system laid the foundation for the evolution of mechanisms conferring tolerance to oxygen.
In addition to all the above corrections, the text has been carefully verified. Academic English has been corrected by a professional translator.
Sincerely,
Authors

Reviewer 2 Report
Comments and Suggestions for Authors
I found this manuscript to be interesting to read. Bacterial communities in soil exposed to radioactive contamination exhibit a wide range of taxonomic diversity and functional characteristics. In general, I think this manuscript of writing is appropriate for publishing, with a few minor remarks taken into consideration.
In Introduction Line no: 49-53 Elaborate on the relevance of the function that the microorganism plays in a soil bacterium.
Explain as to how the radioactive substance will contaminate the earth.
Within the whole of the text, kindly verify the spelling, the size of the front, and the line spacing between the phrases.
Lie no 132-135: Mention a list of any other bacterial strains that have been damaged by the radioactive
Line no 162: How can the amount of irrigating water be increased such that the variety of the soil microorganisms is preserved?
In conclusion: In two lines, please explain the relevance of this work.
Author Response
Dear Reviewer,
Thank you very much for reviewing our manuscript and making critical analysis and valuable remarks. We have revised the entire manuscript as per your suggestions and tried to take all comments into account. All corrections are indicated with a yellow marker. Answers on the specific comments are lower.
I found this manuscript to be interesting to read. Bacterial communities in soil exposed to radioactive contamination exhibit a wide range of taxonomic diversity and functional characteristics. In general, I think this manuscript of writing is appropriate for publishing, with a few minor remarks taken into consideration.
In Introduction Line no: 49-53 Elaborate on the relevance of the function that the microorganism plays in a soil bacterium.
The text was edited as follows: In turn, the activity of microorganisms plays a significant role in influencing the chemical form, mobility, and toxicity of radionuclides (RN) in soils, thereby either impeding or promoting their incorporation into biological migration pathways [39-45]. This is why bacteria together with microscopic fungi are used as a key component in complex biological biostimulants preparations for remediating contaminated soils, particularly in cases of polymetallic and radionuclide pollution [46-48].
Explain as to how the radioactive substance will contaminate the earth.
Thank you for the remark. It seems to us that the introduction contains enough information about the causes of radionuclides entering the soil. More detailed information is provided in the relevant subsections of the article.
Within the whole of the text, kindly verify the spelling, the size of the front, and the line spacing between the phrases.
Thank you for the remark. We have carefully read and edited the text.
Line no 132-135: Mention a list of any other bacterial strains that have been damaged by the radioactive
Thank you for the remark. Unfortunately, we cannot mention specific bacterial strains sensitive to radioactive contamination in the text of our work, because the authors of the works [60] [61] did not identify them.
Line no 162: How can the amount of irrigating water be increased such that the variety of the soil microorganisms is preserved?
Sorry, but it looks like misunderstanding. We mean that bacterial diversity decreases with the dose of radioactivity increase. From our text: The increase in irradiation dosage from artificial radionuclides has led to a reduction in soil bacteria diversity and alterations in their taxonomic composition.
In conclusion: In two lines, please explain the relevance of this work.
Thank you for the remark. The text was added to the following: Thus, the systematization of the data presented in the literature can make a significant contribution to understanding the processes occurring in bacterial communities of radioactively contaminated soils, as well as the main factors affecting the diversity and composition of prokaryotic soil communities during radioactive pollution of their habitat.
In addition to all the above corrections, the text has been carefully verified. Academic English has been corrected by a professional translator.
Sincerely,
Authors

Round 2
Reviewer 1 Report
Comments and Suggestions for Authors
I satisfied with the authors' comments and with the new representation of Figure 1 and ST1. I recommend this manuscript to be published in its present form.